# A Novel Mechanics-Based Design for Overcorrection in Clear Aligner Orthodontics via Finite Element Analysis

**DOI:** 10.3390/bioengineering12020110

**Published:** 2025-01-24

**Authors:** Sensen Yang, Yumin Cheng

**Affiliations:** 1Shanghai Key Laboratory of Mechanics in Energy Engineering, Shanghai Institute of Applied Mathematics and Mechanics, School of Mechanics and Engineering Science, Shanghai University, Shanghai 200072, China; yangsensen@smartee.cn; 2Smartee Biomechanics Research Laboratory, Shanghai Smartee Denti-Technology, Shanghai 201210, China

**Keywords:** clear aligner, overcorrection, finite element analysis, molar distalization

## Abstract

A simplified mechanics model of aligner–tooth interaction was developed to establish a precise computational method for overcorrection design in clear aligner orthodontics. Validated through finite element analysis and experiments, the results demonstrated that designing the movement of only the target teeth on the aligner leads to uneven force distribution on adjacent teeth, while an overcorrection design can evenly distribute the reaction force generated by pushing the target teeth to the anchorage teeth, reducing the maximum force on the anchorage teeth, minimizing unplanned tooth movement, and improving the efficacy of the designed tooth movement for all teeth.

## 1. Introduction

Clear aligners have gained popularity in orthodontics due to their advantages [1]. However, studies report that the average efficacy ranges from 30% to 80%, depending on the type of tooth movement [2,3,4,5,6]. The low efficacy is a result of the mismatch between the intended and realized movement of the aligners and teeth, wherein the actual displacement often falls short of the designed displacement. As treatment progresses, actual tooth movement frequently deviates from the intended trajectory, necessitating refinements [7,8,9,10,11,12]. Overcorrection design has emerged as a common approach to address these efficacy issues in clear aligner orthodontics [13,14,15].

Previous studies on overcorrection design in clear aligner orthodontics have often attempted to provide overcorrection for different angles and displacements, and then evaluated the outcomes [16,17]. However, these studies have lacked a solid theoretical foundation, frequently relying on clinical experience and tentative finite element method calculations [18,19]. Moreover, current research in this area often focuses narrowly on designing overcorrection for individual teeth, without adequately considering the crucial interaction forces between neighboring teeth [20]. In clear aligner orthodontics, the force applied to a particular tooth inevitably arises from the resisting reaction force of other anchorage teeth, with the pushing force and resisting reaction force being equal in magnitude and opposite in direction, as per Newton’s third law [21,22]. Tooth movement occurs through alveolar bone remodeling, a biological response to changes in mechanical stress and strain experienced by the periodontal ligament (PDL) and surrounding tissues [23,24,25]. To move a target tooth while maintaining the position of anchorage teeth, the pushing force must be concentrated on the target tooth, and the resisting reaction force must be evenly distributed across the anchorage teeth. When the PDL stress on the anchorage teeth is below a certain threshold, alveolar bone remodeling does not occur, keeping these teeth stationary [26,27]. Conversely, the target tooth experiences the maximum pushing force and the highest PDL stress levels, triggering alveolar bone remodeling and tooth movement. Since designing the movement of a single tooth on the aligner may lead to excessive forces on adjacent teeth and uneven distribution of anchorage forces, it is necessary to design the displacement of both the target tooth and other teeth on the aligner, rather than merely focusing on the movement of the target tooth alone.

This study presents a novel overcorrection design method that accurately determines the required overcorrection amount by developing a mechanics model of the aligner–tooth interaction. The finite element method is used to validate and refine the model parameters, while a six-axis force/moment testing platform is employed to experimentally verify that pushing a single tooth leads to uneven reactive forces on the anchorage teeth. This approach could maximize the force on target teeth while dispersing and minimizing the maximum force on anchorage teeth, thereby improving overall tooth movement efficacy.

## 2. Materials and Methods

### 2.1. Simplified Mechanics Model

To simplify the analysis of the force distribution between teeth and the aligner, as well as to inform the development of design strategies, the aligner and teeth are simplified into a one-dimensional mechanics model, as illustrated in Figure 1.

Considering left–right symmetry, only half of the model is analyzed. The aligner is represented by six springs connected in series, symbolizing the serial stiffness $K_{2i}(i=1,2,\ldots 6)$ between different teeth positions (Figure 1A). The tooth movement stiffness is denoted by $K_{1i}(i=1,2,\ldots 7)$ (Figure 1B), which approximately represents the ratio of the force exerted by the aligner on the tooth to its maximum displacement. Figure 1C depicts the entire model of the teeth and aligner interacting together, where each tooth is linked to adjacent ones through the aligner.

To further simplify the analysis and calculation, it is assumed that the stiffness between different tooth positions in the aligner model is the same, denoted as $K_{2}$, and the stiffness of all teeth is $K_{1}$. This is because the aligner geometry between adjacent teeth is relatively similar, and the difference in root surface areas among different teeth is not substantial. The key factor is the stiffness ratio between the aligner and the teeth:(1)$$K_{2}=aK_{1}$$

This is only a preliminary assumption, and improvements to this approach are proposed later in this paper.

Assuming that the designed displacement of the aligner is $d_{i}$ and the actual displacement of the teeth is $x_{i}$ (Figure 2), then the actual deformation displacement of the aligner at the tooth position is $d_{i}-x_{i}$, and the corresponding compressive force is $K_{2}\left(d_{i}-x_{i}\right)$. Due to the deformation displacement of the aligner between the preceding tooth and the subsequent tooth being $d_{i}-x_{i}-(d_{i-1}-x_{i-1})$, the compressive forces between different adjacent teeth in the aligner are given by(2)$$F_{12}=K_{2}(d_{2}-x_{2}-(d_{1}-x_{1}))$$(3)$$F_{23}=K_{2}(d_{3}-x_{3}-(d_{2}-x_{2}))$$(4)$$F_{34}=K_{2}(d_{4}-x_{4}-(d_{3}-x_{3}))$$(5)$$F_{45}=K_{2}(d_{5}-x_{5}-(d_{4}-x_{4}))$$(6)$$F_{56}=K_{2}(d_{6}-x_{6}-(d_{5}-x_{5}))$$(7)$$F_{67}=K_{2}(d_{7}-x_{7}-(d_{6}-x_{6}))$$
where the compressive force experienced by different teeth from the PDL is $K_{1}x_{i}$. For each tooth, due to force equilibrium, the sum of the forces exerted by the PDL and the aligner equals zero.(8)$$K_{1}x_{1}-F_{12}=0$$(9)$$F_{12}-K_{1}x_{2}-F_{23}=0$$(10)$$F_{23}-K_{1}x_{3}-F_{34}=0$$(11)$$F_{34}-K_{1}x_{4}-F_{45}=0$$(12)$$F_{45}-K_{1}x_{5}-F_{56}=0$$(13)$$F_{56}-K_{1}x_{6}-F_{67}=0$$(14)$$F_{67}-K_{1}x_{7}=0$$

Substituting Equations (1)–(7) into Equations (8)–(14), upon setting $x_{i}$, the value of $d_{i}$ can be determined.

A simple example is provided below to illustrate the concept and methodology of the overcorrection design. Assuming that the stiffness of the teeth is much greater than the stiffness of the aligner ($K_{1}\gg K_{2}$), the teeth will hardly move under the applied force. Considering only the last four teeth, the objective is to ensure that the first three teeth are subjected to equal forces to push the last tooth. As shown in Figure 3, first, the second molar is moved backward by 3x, resulting in corresponding action and reaction forces of 3F on the first and second molars, respectively. Next, the first and second molars are simultaneously moved backward by 2x, generating an action force of 2F between the second premolar and first molar, while there is no relative displacement between the first molar and second molar, and thus no action force. Finally, the last three teeth are simultaneously moved backward by x, subjecting only the first premolar and second premolar to an action force of magnitude F. Combining these three scenarios together, the resultant force on the second molar is 3F, while the force on each of the first three teeth is F. This achieves the goal of using the first three teeth as anchorage to apply uniform forces to push the last tooth. However, to accomplish this, the designed displacement ratio of the last three teeth should be 1:3:6; the movement design for the first two non-target teeth is an overcorrection design. This indicates that when designing the distalization of molars, designing the movement of only the target molar on the clear aligner is insufficient; incorporating a certain amount of overcorrection for the adjacent teeth is also necessary to truly achieve the objective of solely pushing the molars. In reality, the stiffness difference between the aligner and the teeth is not significant (stiffness ratio < 10). It is necessary to simultaneously consider the influence of both the aligner and tooth displacement on the force system. The following section uses molar distalization as an example to demonstrate this calculation method.

### 2.2. Finite Element Model

The simplified mechanics model is used to develop an overcorrection design strategy, but accurately calculating the overcorrection amount requires determining the stiffness of the clear aligner and teeth through finite element analysis (FEA). The finite element model (Figure 4) comprises the clear aligner, teeth, PDL, and alveolar bone, constructed using Hypermesh (Version 2022, Altair, Troy, MI, USA) and Blender (Version 3.3, Open Source) software. The clear aligner model, with an average thickness of 0.55 mm, is derived from simulating the pressure thermoforming process and uses quadrilateral meshes. Teeth and alveolar bone are modeled as rigid bodies to reduce computational complexity, while the PDL (0.29 mm thickness) is modeled with shared nodes connecting to the alveolar bone and teeth. The friction coefficient between the clear aligner and teeth is 0.16 (as tested by Smartee Laboratory). Table 1 presents the mesh and material parameters. The clear aligner’s positional discrepancy relative to the teeth is resolved using the interference fit method in Abaqus software (Version 2022, Dassault Systemes, Vélizy-Villacoublay, France), and high-precision quadrilateral meshes are employed to ensure the accuracy and convergence of contact calculations.

### 2.3. Clear Aligner Six-Axis Force/Moment Testing Platform

We utilized a clear aligner six-axis force/moment testing system from Smartee (Figure 5) to investigate molar distalization and assess the impact of a single tooth movement on adjacent teeth. The system consists of data acquisition modules, a power supply, computer software (Figure 5A), and a testing platform with 14 six-axis force/moment load cells (SRI, Non-Linearity 1% F.S.), a manual XY linear stage, aluminum bracket, and 14 tooth models (Figure 5B–E). The tooth models are affixed to the load cells, which are mounted on the brackets. One load cell is attached to the manual XY linear stage for precise tooth movement control (0.01 mm resolution). By positioning the clear aligner on the tooth models, the system measures the forces and moments exerted on the teeth in the local xyz directions.

## 3. Results

### 3.1. Simplified Mechanics Model Calculation

It is necessary to obtain the relative stiffness ratio a (Equation (1)) between the aligner and the teeth. The larger the elastic modulus of the aligner, the larger the value of a. Based on the material parameters given in Table 1, a simple finite element case is calculated. By computing the displacements of the aligner and the teeth, it can be found that a is around 3 (a = 3).

A molar distalization case is used as the computational example. For the second molar’s distal movement, the first six teeth serve as anchorage and should be uniformly loaded to push the last tooth. The first six teeth are subjected to equal forces, while the last molar receives the sum of the pushing forces from the six anterior anchorage teeth (Figure 6; Equation (15)).(15)$$K_{1}x_{1}=K_{1}x_{2}=K_{1}x_{3}=K_{1}x_{4}=K_{1}x_{5}=K_{1}x_{6}=-\frac{1}{6}K_{1}x_{7}=F$$

Set the central incisor (tooth #1) as the reference tooth for displacement, with other teeth’s displacements relative to it. Note that if all teeth are designed to move equally on the aligner, it is equivalent to no movement for the teeth, as there is no relative displacement between them. Therefore, the designed displacement of the first tooth is set to 0.0 mm, while the second molar’s designed displacement is assumed to be 0.2 mm.(16)$$d_{1}=0.0;\;d_{7}=0.2$$

Substituting Equations (15) and (16) into Equations (8)–(14) yields the overcorrection design values $d_{i}\;\left(i=2,3,4,5,6\right)$ for the clear aligner, as shown in Table 2.

When designing the clear aligner to solely distalize the second molar, the maximum displacement achieved is 0.113 mm for the second molar and −0.0492 mm for the first molar. However, by applying the corresponding overcorrection design, a uniform displacement of −0.0142 mm is obtained for the first six teeth, with only the second molar experiencing the maximum displacement of 0.0857 mm. This approach allows the first six teeth to evenly distribute the anchorage force, which is then used to distalize the second molar.

Once the second molar has been successfully moved into its desired position, the focus of the treatment shifts to distalizing the first molar. In an ideal overcorrection design for first-molar distalization, the second molar should remain stationary while the pushing force is applied to the first molar. To achieve this, the five anterior teeth should evenly distribute the opposing reaction force among themselves. This allows the first molar to bear the pushing force exerted by the anterior teeth, while the second molar, not subjected to any force, remains stable in its position (Figure 7).(17)$$K_{1}x_{1}=K_{1}x_{2}=K_{1}x_{3}=K_{1}x_{4}=K_{1}x_{5}=-\frac{1}{5}K_{1}x_{6}=F;\;K_{1}x_{7}=0$$(18)$$d_{1}=0.0;\;d_{6}=0.2$$

Similarly, Equations (17) and (18) are obtained. Substituting Equations (17) and (18) into Equations (8)–(14) yields the overcorrection design values for the first molar’s distal movement (Table 3). This design requires a simultaneous 0.109 mm distal movement of the second molar for stability and specific overcorrection values for the four anterior teeth to ensure uniform opposing reaction forces.

### 3.2. Finite Element Analysis

The overcorrection amount for distalizing the first molar has been previously calculated using a simplified model. However, there are certain differences between the simplified model and the actual model. In the following section, we will validate and compare the overcorrection design results using a finite element model of clear aligner orthodontics.

Figure 8A illustrates the finite element model for the design approach involving distalizing only the first molar. The finite element analysis results show that the aligner exhibits a maximum Von Mises stress of 7.627 MPa (Figure 8B). The first molar experiences the maximum displacement (0.1329 mm) and the highest PDL stress (Figure 8C–E; Table 4), while the second molar undergoes the largest opposite displacement (−0.0542 mm) and the second-highest PDL stress. The clear aligner exerts the largest force of −3.257 N on the first molar, followed by 1.2223 N on the second molar (Figure 8F).

Figure 9A illustrates the finite element model for the corresponding overcorrection design. The results show a lower clear aligner maximum Von Mises stress of 5.641 MPa (Figure 9B) compared to the first scheme (7.627 MPa). This leads to a more uniform force distribution among the first five teeth (Figure 9F), with the first molar having the largest displacement (0.0912 mm) and the second molar remaining almost stationary (0.0007 mm) (Figure 9C,G). The PDL’s Von Mises stress and hydrostatic pressure are highest in the first molar and lowest in the second molar (Figure 9D,E; Table 5).

### 3.3. Clear Aligner Six-Axis Force Testing

The effects of moving the second and first molars on the forces experienced by adjacent teeth were examined separately. Figure 10 illustrates the cases where the second and first molars were controlled to move 0.2 mm distally. The forces and moments acting on each tooth in its local xyz coordinate system were obtained and then transformed into the global XYZ coordinate system (Equations (19)–(22)). Table 6 shows the local xyz-direction forces/moments exerted by the clear aligner on teeth #1 to #7 and the XY-direction forces in the global coordinate system. The second molar experienced the maximum force of 2.855 N, while the first molar, closest to the second molar, experienced the second-largest force of −1.203 N. The first six anchorage teeth did not experience uniform forces. When the first molar was controlled to move 0.2 mm distally (Figure 10B), it experienced the largest force of 2.439 N, followed by the second molar with −1.020 N. The forces on the first five teeth were smaller and decreased progressively (Table 7). The test results ($F_{iy}^{\prime }$) closely match the force ratio relationship (Table 6 vs. Table 2; Table 7 vs. Table 3) between different teeth calculated using the simplified model when moving a single tooth.

Due to the current limitations of the experimental platform, which can only move a single tooth at a time, and considering the manufacturing errors of aligners and the manual installation accuracy of model teeth, aligners with an overcorrection design were not directly tested in this study. However, this verifies that directly moving the target tooth in the conventional design is problematic and must be corrected through an overcorrection design.(19)$$Q_{i}^{\prime }=L_{i}Q_{i}$$
wherein(20)$$Q_{i}^{\prime }={\left[\begin{array}{cccccc} F_{ix}^{\prime } & F_{iy}^{\prime } & F_{iz}^{\prime } & M_{ix}^{\prime } & M_{iy}^{\prime } & M_{iz}^{\prime } \end{array}\right]}^{T}$$(21)$$Q_{i}={\left[\begin{array}{cccccc} F_{ix} & F_{iy} & F_{iz} & M_{ix} & M_{iy} & M_{iz} \end{array}\right]}^{T}$$(22)$$L_{i}=\left[\begin{array}{cc} \lambda_{i} & 0\\ 0 & \lambda_{i} \end{array}\right],\;\lambda_{i}=\left[\begin{array}{ccc} \lambda_{x^{\prime }x} & \lambda_{x^{\prime }y} & \lambda_{x^{\prime }z}\\ \lambda_{y^{\prime }x} & \lambda_{y^{\prime }y} & \lambda_{y^{\prime }z}\\ \lambda_{z^{\prime }x} & \lambda_{z^{\prime }y} & \lambda_{z^{\prime }z} \end{array}\right]=\left[\begin{array}{ccc} \mathrm{Cos}\theta_{i} & \mathrm{Sin}\theta_{i} & 0\\ -\mathrm{Sin}\theta_{i} & \mathrm{Cos}\theta_{i} & 0\\ 0 & 0 & 1 \end{array}\right]$$

$Q_{i}^{\prime }$ are the force and moment tensors in the global coordinate system;

$Q_{i}$ are the force and moment tensors in the local coordinate system;

$\lambda_{i}$ are the transformation matrix.

## 4. Discussion

The above finite element analysis and six-axis force/moment platform experiments demonstrate that the simplified mechanics model is effective. The traditional design of independently moving the target tooth on the clear aligner cannot achieve the purpose of solely moving the target tooth. Instead, it will cause the adjacent teeth to experience significant reactive forces, leading to unplanned movement of the neighboring teeth. In the finite element analysis of first-molar distalization (Figure 8), it is observed that besides the target tooth, the second molar experiences the highest force and tends to be pulled mesially, while the second premolar receives the second-highest force and tends to be pushed forward, resulting in unwanted back-and-forth movement of teeth during the distalization process. The six-axis force testing of clear aligners shows that distalizing either the first molar or second molar independently leads to significant forces on adjacent teeth (Table 6 and Table 7), which aligns with the results from both simplified mechanical model calculations and finite element analysis. On the other hand, the finite element analysis results of the overcorrection design for first-molar distalization (Figure 9) demonstrate that the first molar experiences the maximum force with the highest PDL stress, while the second molar receives nearly zero force and remains stationary. The anterior five teeth can more evenly bear the anchorage reaction forces generated by first-molar distalization, thereby reducing the unplanned movement of the anchorage teeth and effectively improving the overall movement efficiency of all teeth.

To clearly explain this mechanics principle, this paper assumes that the teeth have uniform movement stiffness. However, variations in tooth morphology and size in reality lead to inevitable differences in movement stiffness that require correction. As an initial suggestion, it can be proposed that the magnitude of tooth stiffness is linearly related to the size of the root surface area. The following section uses the example of first-molar distalization to calculate the corresponding overcorrection design with root surface area modification. Figure 11 shows the roots of different teeth in this model, and the corresponding root surface areas are listed in Table 8.

The proportional relationship between tooth stiffness and root surface area is obtained as follows:(23)$$\frac{K_{11}}{265.3}=\frac{K_{12}}{266.6}=\frac{K_{13}}{316.6}=\frac{K_{14}}{321.1}=\frac{K_{15}}{254.1}=\frac{K_{16}}{433.5}=\frac{K_{17}}{391.5}$$(24)$$K_{2}=3K_{17}$$(25)$$\frac{K_{11}x_{1}}{265.3}=\frac{K_{12}x_{2}}{266.6}=\frac{K_{13}x_{3}}{316.6}=\frac{K_{14}x_{4}}{321.1}=\frac{K_{15}x_{5}}{254.1}=-\frac{1}{5}\times \frac{K_{16}x_{6}}{433.5};\;K_{17}x_{7}=0$$(26)$$d_{1}=0.0;\;d_{6}=0.2$$(27)$$x_{1}=x_{2}=x_{3}=x_{4}=x_{5}=-\frac{1}{5}x_{6};\;x_{7}=0$$

Similarly, Equations (24)–(26) are obtained. Since the effective stress occurs primarily at the root surface, the force should be proportional to the root surface area. Therefore, Equation (25) is modified to be proportional to the root surface area, resulting in Equation (27). After substituting these into the previous equations, the modified overcorrection design can be obtained (Table 9). It should be noted that in this case, the number of equations exceeds the number of unknowns by one. Typically, we either discard the equation for the tooth furthest from the target tooth, or use the pseudoinverse method to solve the equations. After obtaining the solution through either method, this overcorrection design scheme is then validated using the finite element method, and the results are shown in Figure 12. It can be observed that the anchorage reactive forces experienced by the first five teeth are more evenly distributed compared to the results calculated based on the same stiffness (Table 10 vs. Table 5). The PDL stress patterns (Figure 12E) among anchorage teeth demonstrate greater uniformity and consistency, which better serves the intended biomechanical objective of maintaining anchorage stability while achieving targeted tooth movement. Although this root surface area-based approximation method shows promising results, it is still a simplified approach. If more precise calculations of tooth movement stiffness are required, it is necessary to perform separate finite element analyses for each tooth. While this approach is feasible for research purposes, it is not practical for application to a large number of clinical cases, considering the complexity of creating finite element models and the computational difficulty. In such situations, only approximate methods can be used for rapid calculation while still achieving a good level of accuracy.

Moreover, the accuracy of the finite element model and the precision of the material parameters are crucial for calculating the stiffness coefficients of the simplified model. Many previous models have suffered from accuracy issues, leading to significant computational errors. To address this, our study utilized advanced mesh processing techniques to create a highly accurate finite element model for aligner treatment simulation (including secondary development with Tcl and Python programming, mesh topology segmentation, and hypermorph processing), achieving a positional accuracy within 0.005 mm. Additionally, the use of more accurate material parameters further enhanced the reliability of the simulation results. Existing studies often use inaccurate values, whereas commercially available aligners have elastic moduli ranging from 800 to 2500 MPa. For instance, Invisalign’s SmartTrack has moduli of 1200 MPa (tensile) and 1000 MPa (flexural), while Zendura FLX by Bay Materials has 1000 MPa (tensile) and 1600 MPa (flexural). Smartee provides a range of aligner sheets with elastic moduli varying from 800 to 1600 MPa. Our research has found that aligner materials with an elastic modulus below 600 MPa are too soft and not suitable for the manufacturing of clear aligners. Therefore, we adopted an elastic modulus of 1200 MPa, which is relatively close to the material parameters of most aligners currently in use.

Another crucial factor is the mechanical properties of the PDL. Although the PDL is generally considered a non-linear elastic material [28,29,30,31], it can be approximated as a linear elastic material when loading history is not considered. To study the PDL’s mechanical properties, we constructed a human tooth biomechanical testing apparatus (Figure 13A,B) that measures the force–displacement relationship of human teeth (Figure 13C). By combining experimental data with finite element method inverse calculations, the PDL’s elastic modulus can be determined [32,33]. Our FEA and experimental research revealed that when the PDL’s elastic modulus is significantly greater than 0.1 MPa, it substantially increases the force on teeth and stress levels within the PDL. Due to the likely higher tooth stiffness in the six-axis force/moment testing platform compared to the finite element model, the finite element results should be lower than those from the six-axis force testing. However, the opposite is observed, suggesting that the PDL’s elastic modulus is probably less than 0.1 MPa. Thus, future research on the PDL’s mechanical properties, varying thickness and morphology will be a key focus area.

Although this article presents a simplified one-dimensional mechanics model and provides an example of molar distalization, which differs from actual three-dimensional tooth movement, the principles can still be broadly applied to all teeth and any type of tooth movement in three dimensions. For instance, the overcorrection design for controlling tooth rotation can be represented by rotational stiffness in the equation separately in three directions and then superimposed to obtain the overall effect.

In the stepwise design of tooth movement, multiple teeth are moved simultaneously at each stage. An overcorrection design can be applied to the incremental displacement of each individual tooth. By sequentially accumulating the overcorrection values for each tooth, the total overcorrection design for all teeth in that particular stage can be determined.

## 5. Conclusions

The traditional aligner design approach, focusing solely on moving target teeth, can lead to the unintended movement of adjacent teeth due to uneven force distribution. To address this issue, we developed a simplified mechanics model of the aligner–tooth interaction and proposed an overcorrection design method. Our approach evenly distributes the reaction force generated by pushing the target teeth to the anchorage teeth, minimizing unplanned tooth movement and improving the efficacy of the designed tooth movement for all teeth.

## Figures and Tables

**Figure 1 bioengineering-12-00110-f001:**
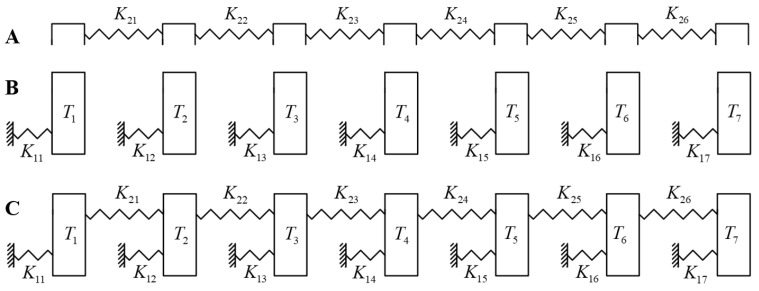
(**A**) Simplified mechanics model of clear aligner; (**B**) simplified mechanics model of teeth; (**C**) entire model of interaction between clear aligner and teeth.

**Figure 2 bioengineering-12-00110-f002:**
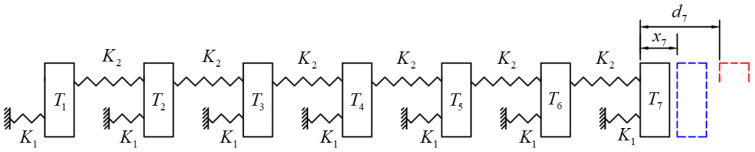
The tooth movement design of the clear aligner ($d_{i}$) and the actual tooth displacement ($x_{i}$).

**Figure 3 bioengineering-12-00110-f003:**
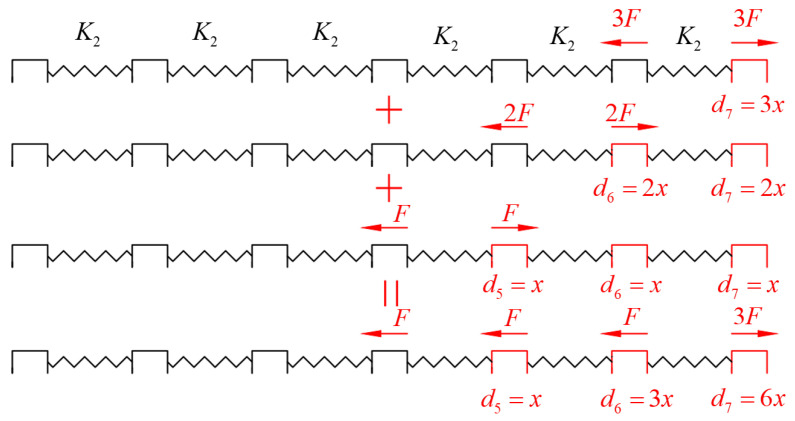
A simple example of the overcorrection design for second-molar distalization.

**Figure 4 bioengineering-12-00110-f004:**
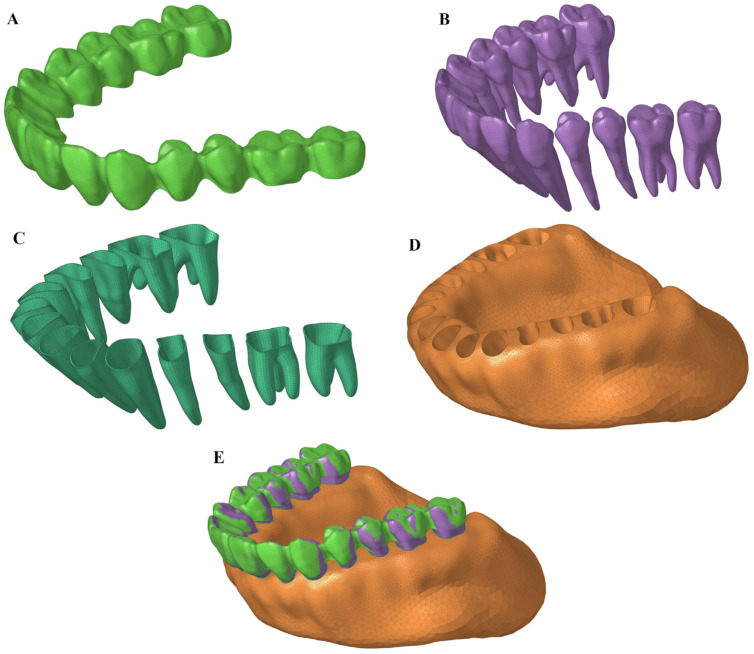
(**A**) Clear aligner (CA); (**B**) teeth; (**C**) PDL; (**D**) alveolar bone; (**E**) the entire model.

**Figure 5 bioengineering-12-00110-f005:**
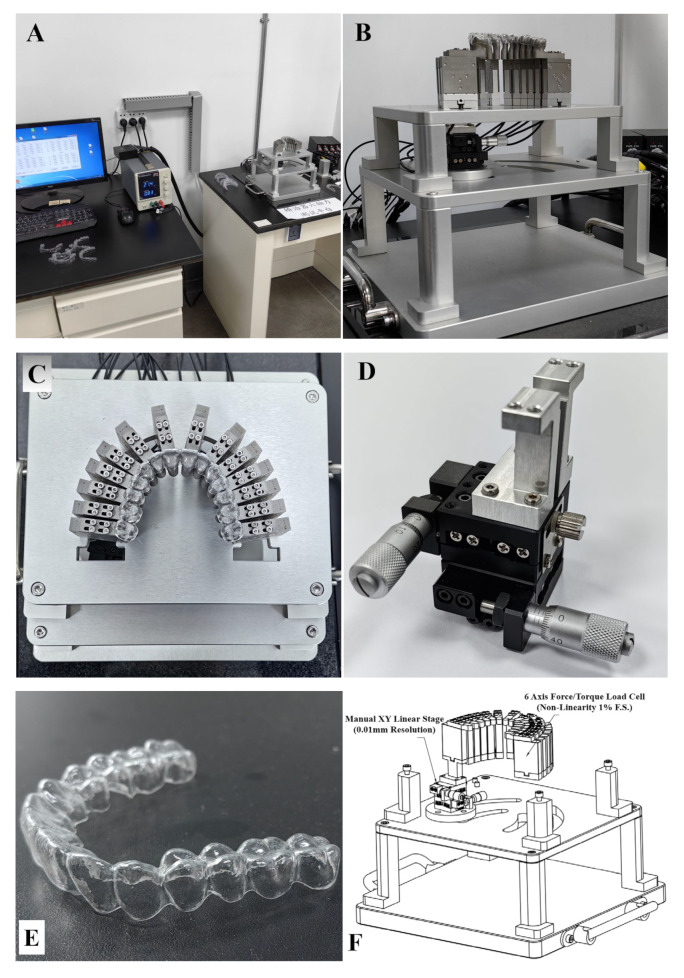
Clear aligner six-axis force/moment testing system: (**A**) data acquisition modules, power supply and computer software; (**B**) clear aligner six-axis force/moment testing platform; (**C**) top view of the testing platform; (**D**) manual XY linear stage; (**E**) the test clear aligner sample; (**F**) structure diagram of the testing platform.

**Figure 6 bioengineering-12-00110-f006:**
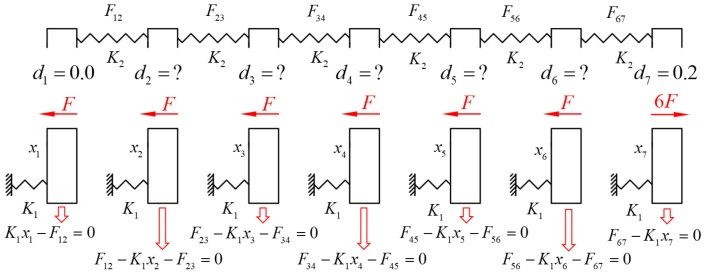
The mechanics model for the second-molar distalization.

**Figure 7 bioengineering-12-00110-f007:**
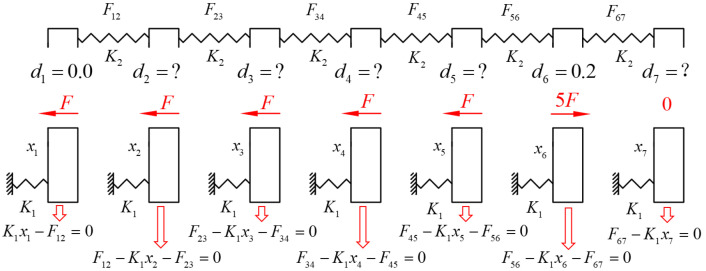
The mechanics model for the first-molar distalization.

**Figure 8 bioengineering-12-00110-f008:**
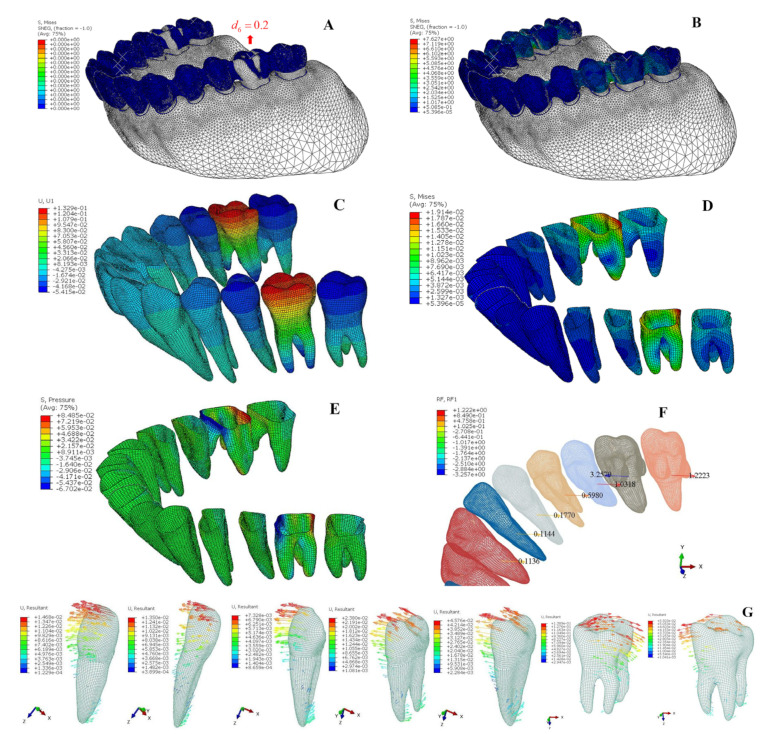
(**A**) The initial state of the distalization of 0.2 mm for the first molar in a clear aligner; (**B**) the FEA results of interference fit calculation between the clear aligner and the teeth; (**C**) tooth displacement in the x-direction (mm); (**D**) Von Mises stress for the PDL (MPa); (**E**) hydrostatic pressure for the PDL (MPa); (**F**) reaction force at teeth root (N); (**G**) the displacement vectorgraph of each tooth (mm).

**Figure 9 bioengineering-12-00110-f009:**
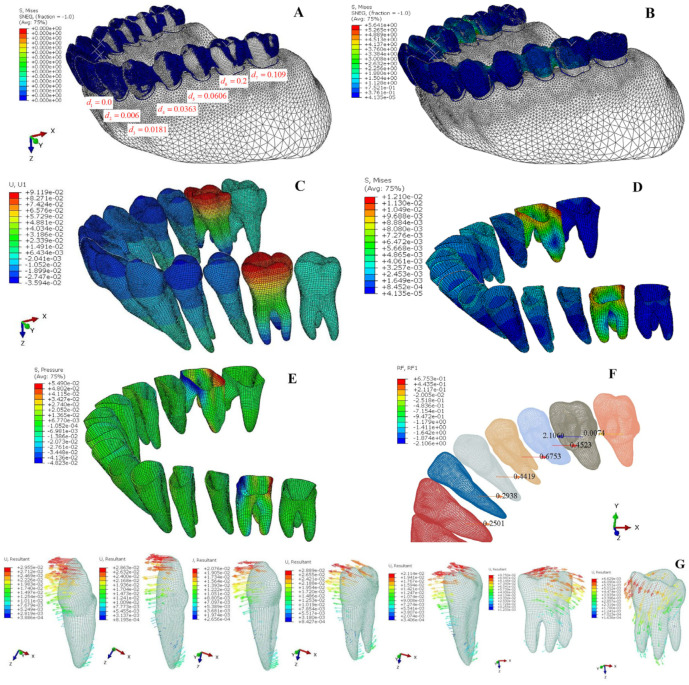
(**A**) The initial state of the overcorrection design in the clear aligner; (**B**) the FEA results of the interference fit calculation between the clear aligner and the teeth; (**C**) tooth displacement in the x-direction (mm); (**D**) Von Mises stress for the PDL (MPa); (**E**) hydrostatic pressure for the PDL (MPa); (**F**) reaction force (x-dir) at teeth root (N); (**G**) the displacement vectorgraph of each tooth (mm).

**Figure 10 bioengineering-12-00110-f010:**
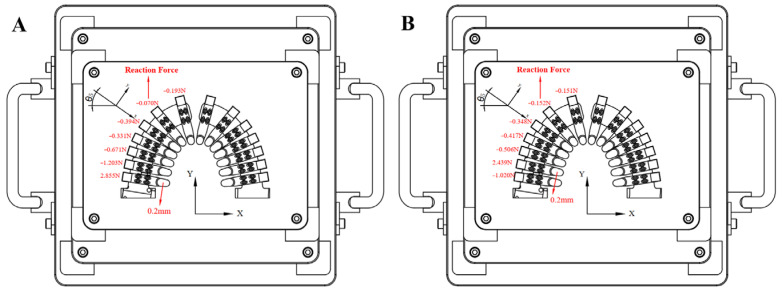
(**A**) Moving the second molar back 0.2 mm; (**B**) moving the first molar back 0.2 mm.

**Figure 11 bioengineering-12-00110-f011:**
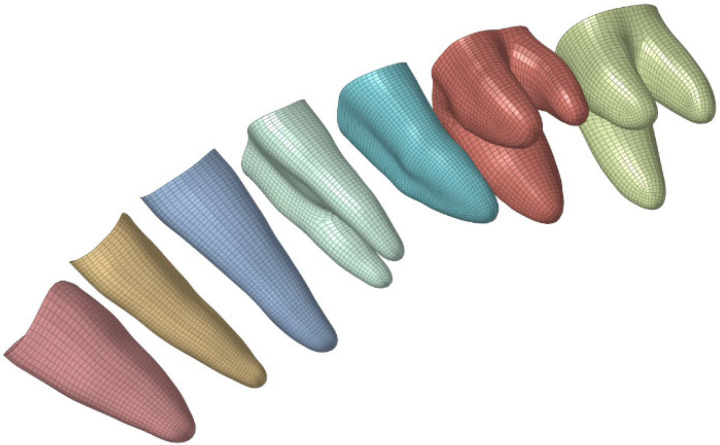
Teeth root.

**Figure 12 bioengineering-12-00110-f012:**
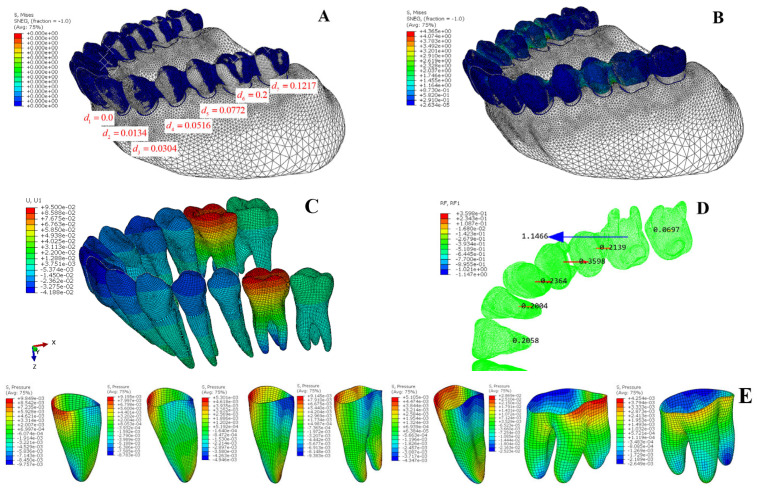
FEA results of overcorrection design based on root surface area adjustment: (**A**) the initial state of the overcorrection design in the clear aligner; (**B**) the FEA results of the interference fit calculation between the clear aligner and the teeth; (**C**) tooth displacement in the x-direction (mm); (**D**) reaction force (x-dir) at teeth root (N); (**E**) hydrostatic pressure distribution in the PDL of each tooth (MPa).

**Figure 13 bioengineering-12-00110-f013:**
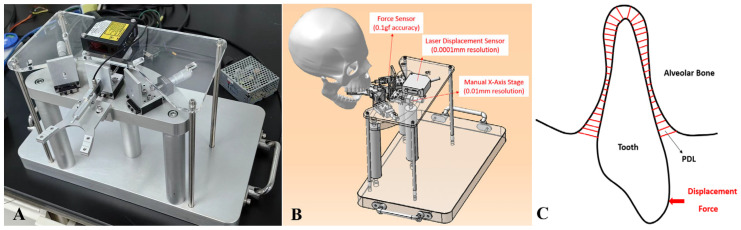
(**A**) Human tooth biomechanical testing apparatus; (**B**) configuration of the testing apparatus; (**C**) schematic diagram of tooth force–displacement test.

**Table 1 bioengineering-12-00110-t001:** The FEA model parameters (half of the show model).

Model	Elements	Nodes	Element Type	Elastic Modulus (MPa)	Poisson’s Ratio
CA	45,030	68,153	S4R	1200	0.46
PDL	17,608	58,275	C3D8R	0.1	0.45
Teeth	35,694	58,286	S4R (99.5%) + S3	Rigid body	Rigid body
Alveolar Bone	27,408	45,175	S4R (64.2%) + S3	Rigid body	Rigid body

**Table 2 bioengineering-12-00110-t002:** The distalization of 0.2 mm for the second-molar design and its overcorrection design.

Teeth #	1	2	3	4	5	6	7
Initial Design ($d_{i}$)	0.0	0.0	0.0	0.0	0.0	0.0	0.2
Displacement of Teeth ($x_{i}$)	−0.0044	−0.0059	−0.0094	−0.0160	−0.0279	−0.0492	0.1130
Overcorrection Design ($d_{i}$)	0.0	0.0047	0.0142	0.0285	0.0476	0.0714	0.2
Displacement of Teeth ($x_{i}$)	−0.0142	−0.0142	−0.0142	−0.0142	−0.0142	−0.0142	0.0857

**Table 3 bioengineering-12-00110-t003:** The distalization of 0.2 mm for the first-molar design and its overcorrection design.

Teeth #	1	2	3	4	5	6	7
Initial Design ($d_{i}$)	0.0	0.0	0.0	0.0	0.0	0.2	0.0
Displacement of Teeth ($x_{i}$)	−0.0059	−0.0079	−0.0125	−0.0213	−0.0372	0.1343	−0.0492
Overcorrection Design ($d_{i}$)	0.0	0.0060	0.0181	0.0363	0.0606	0.2	0.1090
Displacement of Teeth ($x_{i}$)	−0.0181	−0.0181	−0.0181	−0.0181	−0.0181	0.0909	0.0

**Table 4 bioengineering-12-00110-t004:** The FEA results for the design of 0.2 mm distalization of the first molar.

Teeth #	1	2	3	4	5	6	7
Initial design (mm)	0.0	0.0	0.0	0.0	0.0	0.2	0.0
U1(Max|U1|) (mm)	−0.0130	−0.0114	−0.0070	−0.0237	−0.0455	0.1329	−0.0542
Max Mises of PDL (MPa)	0.0015	0.0015	0.0010	0.0031	0.0059	0.0191	0.0066
Max/Min hydrostatic pressure of PDL (MPa)	0.0062/−0.0061	0.0065/−0.0064	0.0035/−0.0039	0.0126/−0.0138	0.0257/−0.0225	0.0848/−0.0670	0.0279/−0.0308
Reaction force at x-dir (N)	0.1136	0.1144	0.1770	0.5980	1.0318	−3.2570	1.2223

**Table 5 bioengineering-12-00110-t005:** The FEA results for the overcorrection design of 0.2 mm distalization of the first molar.

Teeth #	1	2	3	4	5	6	7
Overcorrection design	0.0	0.0060	0.0181	0.0363	0.0606	0.2	0.1090
U1(Max|U1|) (mm)	−0.0264	−0.0249	−0.0195	−0.0285	−0.0203	0.0912	0.0007
Max Mises of PDL (MPa)	0.0030	0.0032	0.0024	0.0038	0.0028	0.0121	0.0009
Max/Min hydrostatic pressure of PDL (MPa)	0.0127/−0.0123	0.0140/−0.0139	0.0112/−0.0100	0.0159/−0.0171	0.0111/−0.0105	0.0549/−0.0482	0.0043/−0.0029
Reaction force at x-dir (N)	0.2501	0.2938	0.4419	0.6753	0.4523	−2.1060	−0.0074

**Table 6 bioengineering-12-00110-t006:** Moving the second molar back 0.2 mm (y-dir) and its reaction force from the clear aligner.

Teeth #	1	2	3	4	5	6	7
$\theta_{i}$ (Deg)	75.54	50.312	35.17	24.832	17.305	11.733	8.703
$F_{ix}$ (N)	0.1644	−0.0132	0.3427	0.1388	0.0409	−0.2158	0.508
$F_{iy}$ (N)	−0.1388	−0.1267	−0.2407	−0.301	−0.6911	−1.2735	2.9667
$F_{iz}$ (N)	−0.0111	0.0066	0.1628	−0.0364	−0.3003	−0.8978	0.8971
$M_{ix}$ (Nm)	0.0008	0.0000	0.0008	−0.0001	0.0005	0.0007	0.0005
$M_{iy}$ (Nm)	0.0031	0.0014	0.0009	−0.0011	0.0006	0.0019	−0.0005
$M_{iz}$ (Nm)	−0.0003	−0.0004	−0.0013	−0.0004	−0.0014	−0.0084	0.0084
$F_{ix}^{\prime }$ (N)	−0.0933	−0.1059	0.14149	−0.0004	−0.1665	−0.4702	0.9510
$F_{iy}^{\prime }$ (N)	−0.1938	−0.0707	−0.3941	−0.3314	−0.6719	−1.2030	2.85567

**Table 7 bioengineering-12-00110-t007:** Moving the first molar back 0.2 mm (y-dir) and its reaction force from the clear aligner.

Teeth #	1	2	3	4	5	6	7
$F_{ix}$ (N)	0.1352	0.0845	0.2557	0.1366	0.0028	0.2991	−0.1135
$F_{iy}$ (N)	−0.0805	−0.1361	−0.2451	−0.3961	−0.5296	2.5528	−1.0494
$F_{iz}$ (N)	0.0488	−0.0357	0.0756	0.0721	−0.7218	0.4144	0.1858
$M_{ix}$ (Nm)	0.0008	−0.0002	0.0012	−0.0004	0.0024	−0.0018	−0.0004
$M_{iy}$ (Nm)	0.0019	0	0.0003	0.002	0.0033	−0.0064	0.0013
$M_{iz}$ (Nm)	−0.0003	−0.0006	0.0004	−0.0002	0.0002	0.0087	−0.0049
$F_{ix}^{\prime }$ (N)	−0.0441	−0.0507	0.0678	−0.0423	−0.1548	0.8119	−0.2709
$F_{iy}^{\prime }$ (N)	−0.1510	−0.1519	−0.3476	−0.4168	−0.5064	2.4386	−1.0201

**Table 8 bioengineering-12-00110-t008:** Root surface area.

Tooth #	1	2	3	4	5	6	7
Root surface area (${\mathrm{mm}}^{2}$)	265.3	266.6	316.6	321.1	254.1	433.5	391.5

**Table 9 bioengineering-12-00110-t009:** Results of simplified mechanics model for overcorrection design based on root surface area adjustment.

Teeth #	1	2	3	4	5	6	7
Overcorrection Design ($d_{i}$)	0.0	0.0134	0.0304	0.0516	0.0772	0.2	0.1217
Displacement of Teeth ($x_{i}$)	−0.0156	−0.0156	−0.0156	−0.0156	−0.0156	0.0782	0.0

**Table 10 bioengineering-12-00110-t010:** The FEA results of the overcorrection design.

Teeth #	1	2	3	4	5	6	7
Overcorrection Design	0.0	0.0134	0.0304	0.0516	0.0772	0.2	0.1217
U1(Max|U1|) (mm)	−0.0418	−0.0333	−0.0189	−0.0305	−0.0180	0.0950	0.0069
Max/Min Hydrostatic Pressure of PDL (MPa)	0.0098/−0.0097	0.0091/−0.0087	0.0053/−0.0049	0.0091/−0.0093	0.0051/−0.0043	0.0286/−0.0252	0.0042/−0.0026
Reaction Force at x-dir (N)	0.2058	0.2004	0.2364	0.3598	0.2139	−1.1466	−0.0697

## Data Availability

The original contributions presented in this study are included in the article. Further inquiries can be directed to the corresponding author(s).

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
