# Peer review of "A Novel Mechanics-Based Design for Overcorrection in Clear Aligner Orthodontics via Finite Element Analysis"

_bioengineering, 2025, doi:10.3390/bioengineering12020110_

Round 1

Reviewer 1 Report

Comments and Suggestions for Authors

This is a well designed lab based study. It will be helpful for the readers with n knowledge of finite analysis element studies, if the authors may include a paragraph in the discussion explaining the results in layperson terms

Author Response

Comment:

This is a well designed lab based study. It will be helpful for the readers with n knowledge of finite analysis element studies, if the authors may include a paragraph in the discussion explaining the results in layperson terms

Response:

We thank the reviewer for his comments. In the revised paper, we have added the discussion of the finite element analysis results for better clarity.

Reviewer 2 Report

Comments and Suggestions for Authors

Thanks for inviting me to review this paper. Although I am an Orthodontist, I did my best to follow the procedures mentioned in this manuscript's Materials and Methods section.

The manuscript is good, and I would like to congratulate the authors on their efforts in producing this piece of research work. However, I have these small points that require addressing.

Title

1- The validation procedure of the proposed method should be mentioned in the title, i.e., the use of the FEA model to prove the superiority of the given equations. Therefore, I suggest modifying your title a little bit.

Introduction

2- In the second paragraph, the first three sentences require supporting by appropriate references. One citation is needed for each sentence of these three.

3- In Line 46, you mentioned, "To achieve this selective orthodontic effect, it is necessary to design the displacement of both the target tooth and other teeth on the aligner, rather than merely focusing on the movement of the target tooth alone."; for where did you take this assumption?\

Materials and Methods

4- In Lines 74 to 76, you mentioned, "This is because the aligner stiffness between the tooth positions is relatively similar, while the stiffness of the teeth is mainly related to the morphology and size of the tooth roots, which generally does not exceed a twofold difference"; from where did you take this ratio?

Results

Fine

Discussion 

Fine

Conclusions

Fine.

Good Luck

Author Response

Comment:

1- The validation procedure of the proposed method should be mentioned in the title, i.e., the use of the FEA model to prove the superiority of the given equations. Therefore, I suggest modifying your title a little bit.

Response:

We thank the reviewer for his comments. And we have modified the title to "A Novel Mechanics-Based Design for Overcorrection in Clear Aligner Orthodontics via Finite Element Analysis" to better reflect the finite element analysis methodology.

Comment:

2- In the second paragraph, the first three sentences require supporting by appropriate references. One citation is needed for each sentence of these three.

Response:

We added references [16, 17], [18, 19], and [20] respectively to these three sentences in sequence.

Comment:

3- In Line 46, you mentioned, "To achieve this selective orthodontic effect, it is necessary to design the displacement of both the target tooth and other teeth on the aligner, rather than merely focusing on the movement of the target tooth alone."; for where did you take this assumption?

Response:

This assumption is based on our own experimental findings and finite element analysis, without prior literature support, which demonstrate that designing the movement of a single tooth on the aligner may lead to excessive forces on adjacent teeth and uneven distribution of anchorage forces. Conversely, designing corresponding displacements for adjacent teeth will definitely correct this effect, resulting in an even distribution of anchorage forces. We have revised the sentence accordingly.

Comment:

4- In Lines 74 to 76, you mentioned, "This is because the aligner stiffness between the tooth positions is relatively similar, while the stiffness of the teeth is mainly related to the morphology and size of the tooth roots, which generally does not exceed a twofold difference"; from where did you take this ratio?

Response:

We have removed the specific ratio statement since it was a preliminary assumption based on general tooth root morphology differences, although exceptions may exist. We have revised the sentence accordingly.